# Improving the Named Entity Recognition of Chinese Electronic Medical Records by Combining Domain Dictionary and Rules

**DOI:** 10.3390/ijerph17082687

**Published:** 2020-04-14

**Authors:** Xianglong Chen, Chunping Ouyang, Yongbin Liu, Yi Bu

**Affiliations:** 1School of Computer, University of South China, Hengyang 421001, China; dragonc.cxl@gmail.com (X.C.); yongbinliu03@gmail.com (Y.L.); 2Center for Complex Networks and Systems Research, Luddy School of Informatics, Computing, and Engineering, Indiana University, Bloomington, IN 47408, USA; buyi@umail.iu.edu

**Keywords:** entity recognition, electronic medical records, Bi-LSTM-CRF, rules, domain dictionary

## Abstract

Electronic medical records are an integral part of medical texts. Entity recognition of electronic medical records has triggered many studies that propose many entity extraction methods. In this paper, an entity extraction model is proposed to extract entities from Chinese Electronic Medical Records (CEMR). In the input layer of the model, we use word embedding and dictionary features embedding as input vectors, where word embedding consists of a character representation and a word representation. Then, the input vectors are fed to the bidirectional long short-term memory to capture contextual features. Finally, a conditional random field is employed to capture dependencies between neighboring tags. We performed experiments on body classification task, and the F1 values reached 90.65%. We also performed experiments on anatomic region recognition task, and the F1 values reached 93.89%. On both tasks, our model had higher performance than state-of-the-art models, such as Bi-LSTM-CRF, Bi-LSTM-Attention, and Vote. Through experiments, our model has a good effect when dealing with small frequency entities and unknown entities; with a small training dataset, our method showed 2–4% improvement on F1 value compared to the basic Bi-LSTM-CRF models. Additionally, on anatomic region recognition task, besides using our proposed entity extraction model, 12 rules we designed and domain dictionary were adopted. Then, in this task, the weighted F1 value of the three specific entities extraction reached 84.36%.

## 1. Introduction

Named entity recognition (NER) in biomedicine is an important part of building a biomedical information database. It is also a prerequisite for tasks such as entity relationship extraction. With the rapid development of medical information, many electronic medical records (EMR) are increasingly accumulating. EMR contain a large amount of medical knowledge and patient health information [1]. However, EMR usually contain text, charts, and images. These unstructured data are often irregular or incomplete, and it is difficult to extract information from unstructured data directly. Specifically, to convert EMR data into standardized, structured data, it is critical to study effective ways of named entity recognition of EMR. The main aim of NER is to identify entities in EMR, such as body parts, diseases, primary tumor site, lesion size, and transfer site. These medical entities are very important to follow-up research, such as medical knowledge graph building, medical intelligent QA (Question and Answer), and intelligent medical-aided diagnostic.

In recent years, EMR named entity recognition has attracted a large amount of research and many entity extraction methods have been proposed [2,3,4]. Traditional methods are either rule-based [5] or dictionary-based [6], which usually need to be formulated manually and only for the current dataset. When the dataset is replaced, it leads to high system resources cost to replace the rules. If the original rules or dictionary are used, it will cause low recall. The current advanced approaches are data-driven, including machine learning methods [7] and deep learning methods, especially the Bidirectional Long Short-Term Memory and Conditional Random Field (Bi-LSTM-CRF) models [8,9,10,11], which have been successfully used, achieved better results, and represent some of the more commonly used methods. Despite the great success of the deep learning method, there are still some problems that have not been well resolved. A major drawback is that this approach rarely considers the integration of human knowledge.

Previous studies on clinical NER have primarily focused on English. Compared to the generic domain, NER in medical domain is more difficult for two main reasons: First, many entities rarely or even do not occur in the training set because of the use of non-standard abbreviations, acronyms, jargons, and multiple variations of same entities. Second, notes in EMR are noisy, containing grammatical errors and less context due to shorter and incomplete sentences. More seriously, clinical NER in Chinese is more difficult compared with that in English. On the one hand, there is no word boundary in Chinese, which makes it difficult to recognize composite entities. For example, in ‘the right hilar lymph node’, ‘right hilum’ can describe a site, while ‘the lymph node’ can also describe a site. On the other hand, researchers cannot easily access the EMR resources, as they contain private user data, which leads to data-driven methods having difficulty achieving better results. In some specific tasks, it is not necessary to find all entities in EMR, thus an entity extraction model cannot complete such tasks.

To resolve the various problems about NER in Chinese electronic medical records (CEMR), in this paper, we improve Bi-LSTM-CRF [12] to model the NER of CEMR task as a sequential marking problem and use dictionary features to help identify rare and unseen clinical named entities. Specifically, given a sequence of words, our model first represents each word using a low-dimensional vector concatenated from its corresponding word and character sequence embeddings. Then, the word vector and dictionary features are projected into dense vector representations. They are then fed to the Bidirectional Long Short-Term Memory (Bi-LSTM) to capture contextual features. Finally, a Conditional Random Field (CRF) is employed to capture dependencies between neighboring tags. We made extensive comparisons of the entity extraction models using the 2017 China Conference on Knowledge Graph and Semantic Computing Task 2 (CCKS-2017 Task 2) and the 4th China Health Information Processing Conference Task 1 (CHIP-2018 Task 1) datasets, achieving highly competitive performance. Under specific task requirements, we used the proposed entity extraction model, combined with rules and external dictionaries, to achieve good results on the CHIP-2018 dataset.

The main contributions of this paper are as follows: Firstly, we incorporate characters, words, and dictionary features into the entity extraction model and use it for the NER task of EMR. The effectiveness of the method was proved by experiments. In particular, our model has a good effect on dealing with small frequency entities and unknown entities; with a small training dataset, our method had an average F1 value of 4.91% higher than the basic Bi-LSTM-CRF. Secondly, in the CHIP 2018 task, using our proposed entity extraction model combined with rules and dictionaries, we achieved 84% on F1 value. Because of the interaction between rules, there is no linear relationship between the increase of rules and the experimental results.

Section 2 presents a survey of related work on the subject. Section 3 explains the proposed technique in depth. Section 4 presents the experiments and results. Section 5 discusses several important issues of the proposed model. Finally, Section 6 concludes the current paper and points out potential future work.

## 2. Related Work

NER has a long history in the field of natural language processing (NLP), which aims at automatic detection of named entities (e.g., person, location, disease, and medicine). Early approaches to NER typically use rule-based and dictionary-based methods. Recently, most approaches are data-driven.

Rule-based methods rely on hand-crafted features such as syntactic features [13] and word length [14]. However, it is difficult to list all the rules to solve the named entity task; thus, the manual rule method will result in higher system engineering costs and, when adding a new rule, it is hard to know if there is a conflict with the previous rules. Dictionary-based methods rely on existing domain vocabularies to identify entities and they are widely used due to their simplicity and performance [6,15]. The dictionary needs to be built manually. The entity recognition of electronic medical records can be realized by combining dictionaries in the clinical field. However, building a complete dictionary is difficult and these methods do not recognize entities that do not appear in the dictionary.

Machine learning and deep learning are data-driven methods. They consider NER task as a sequence labeling problem where the goal is to find the best label sequence for a given input sentence. Previous approaches have used sequence labeling models such as hidden Markov models (HMM) [16], support vector machines (SVM) [17,18], maximum entropy Markov models (MEMM) [7], and conditional random fields (CRF) [19,20,21]. These machine learning methods, however, all rely on feature engineering. According to different datasets, it is necessary to find the best feature set that helps to distinguish the entities, which makes the development cost higher. The generalization ability of these methods is not very satisfying; often, the same rules that have achieved good results on the training set are not effective on the test set.

In recent years, deep learning has performed well in many areas of natural language processing, such as part-of-speech tagging, text segmentation, intelligent question-answering, and sentiment analysis. It becomes more popular to use deep learning methods to study named entity recognition [2,22,23,24,25]. Fu et al. [26] used graph convolutional networks to jointly learn entity recognition and relation extraction. At the same time, many scholars apply active learning [27], transfer learning [8,28], and other methods to entity recognition task. Because there are many difficulties in Chinese entity recognition, it attracts a lot of researchers. Ouyang et al. [29] adopted Bidirectional Recurrent Neural Networks and Conditional Random Field (Bi-RNN-CRF) architecture with concatenated n-gram character representation to recognize Chinese clinical named entities. They also incorporated word segmentation results, part-of-speech tagging, and medical vocabulary as features into their model. Hu et al. [30] developed a hybrid system based on rules, CRF, and LSTM methods for the Chinese named entity recognition (CNER) task. They also utilized a self-training algorithm on extra unlabeled clinical texts to improve recognition performance. Zhou et al. [31] proposed a two-way LSTM neural network model based on attention mechanism for relation extraction research. The attention mechanism can automatically find words that play a key role in classification, so that this model can capture the most important semantic information from each sentence.

In the past years, the entity recognition systems in the medical domain usually used the main machine learning algorithms: SVM, HMM, CRF, structured support vector machine (SSVM), etc. [32]. Because deep learning has shown great potential in natural language processing, some researchers used deep learning to address some problems of medical NER. Pyysalo et al. [33] used word2vec to train a list of medical resources, and obtained a better performance on disease NER. Recently, Xu et al. proposed a hybrid model named Semantics Bidirectional LSTM and CRF (SBLC) for disease NER [34] and achieved a good F1 score. However, few studies apply deep learning models to medical NER. Motivated by the effectiveness of both applied LSTM model and embeddings, we propose a hybrid model combining the LSTM model, CRF model, and word embedding, which integrates character, word, and dictionary features to improve input data quality for NER of CEMR.

## 3. Method

In this work, a hybrid entity extraction model is proposed to extract entities from CEMR. Figure 1 shows the overview of our model. The proposed model consists of three parts: vector representation, entity extraction model, and rules for specific tasks. Specifically: (1) Vector representation provides input data for the entity extraction model, consisting of the word vector (built as a combination of character embedding and word embedding) and the dictionary feature vector. (2) The entity extraction model first obtains the context features by taking the vector representation as the input of the bidirectional LSTM, and then decodes the tag sequence globally through the CRF layer. (3) Rules for specific task part includes some manual definition rules from selecting sentences and external dictionaries for some specific tasks. In this part, the rules could update to meet the requirements of different tasks.

### 3.1. Vector Representation

The named entity recognition of CEMR is usually regarded as a sequence labeling task. Given an EMR sentence $X=\left(x_{1},\cdots ,x_{n}\right)$, our goal is to label each word $x_{i}$ in the sentence X with BIEOS (Begin, Inside, End, Outside, Single) tag scheme and obtain a tag sequence $Y=\left(y_{1},\cdots ,y_{n}\right)$. An example of the tag sentence for ‘Lymph node enlargement in the right breast area, the change is not significant’ is shown in Table 1. The commonly used annotation modes for training data of named entity recognition are BIO, BIEO, and BIESO, where B represents the beginning of entity, I represent the middle of entity, E represents the end of entity, S represents a single entity, and O represents not an entity. Depending on the dataset, the types of entities are different; thus, after each label, the abbreviation of entities is added to determine the types of entities. For example, body parts are represented as b, and B-b is the beginning of body parts.

Due to the ambiguity in the boundary of Chinese words and the difficulty of processing rare and unseen entities well, in this work, we construct a new vector representation by combining dictionary feature vector. The vector representation $e_{i}$ of each word $x_{i}$ consists of the word vector $w_{i}$ and the dictionary feature vector $d_{i}$ as $e_{i}=w_{i}\;⊕\;d_{i}$, where ⊕ is the concatenating operator.

Word to vector: As developed by Mikolov [35], each word $x_{i}$ is represented by a word vector $w_{i}$, and the purpose is to facilitate calculations. The word vector is built as a combination of word embedding and character embedding, as illustrated by Figure 2. We use bidirectional LSTM to extract character features, and adopt these features to represent character embedding. Although the computational cost of LSTM is much higher than CNNs (details about this can be found in [36]), LSTM recurrent neural network slightly outperforms CNN as a character level encoder. Word embedding is trained by word2vec [35]. We use highway network to connect Character embedding $w_{i}^{char}$ and word embedding $w_{i}^{word}$, which can retain some independence of word and character when merging with interaction:(1)$$t=\sigma \left(W_{t}\left(w_{i}^{char},w_{i}^{word}\right)+b_{t}\right)$$
(2)$$w_{i}=t\;⊙\;g\left(W_{k}\left(w_{i}^{char},w_{i}^{word}\right)+b_{k}\right)+\left(1-t\right)\;⊙\;\left(w_{i}^{char},w_{i}^{word}\right)$$
where g is a nonlinear function tanh, t is a gate control mechanism, $W_{t}$ and $W_{k}$ are square weight matrices, and $b_{t}$ and $b_{k}$ are bias vectors, respectively.

Dictionary feature to vector: Firstly, we divide the original sentence into text segments based on the context of word $x_{i}$ using n-gram templates (see Table 2 for details). After that, combined with domain dictionary D, we generate a binary value based on whether the text segments are in D or not. According to the number of entity categories in the dictionary, we construct dictionary feature vectors in different dimensions. In our method, the dictionary contains a class of entities (body). Due to the number of templates, the dictionary feature is represented by a seven-dimensional vector. The final dictionary feature vector $d_{i}$ is converted from dictionary feature by Bi-LSTM.

### 3.2. Entity Extraction Model

In this section, we describe the entity extraction model of CEMR. The entity extraction model adopted in this paper uses Bi-LSTM-CRF based on the idea of Huang et al. [2]. However, different from Huang et al., who simply employed word embedding as inputs, we employ character embedding, word embedding, and dictionary feature vector together. This helps obtain more feature information in practice.

The overall process of entity extraction model is as follows: Firstly, in the input of the model, we fuse character, words, and dictionary feature vectors. This has the following advantages: (1) more feature information; and (2) better determination of the entity boundary problem through the dictionary features. Second, we feed this into a Bi-LSTM model. Instead of separating the three feature vectors as inputs, the hidden state *h* (consisting of three hidden states) is obtained by three independent Bi-LSTM models [4]. The general architecture of the proposed model is illustrated in Figure 3. 

LSTM Layer: The recurrent neural network (RNN) is an artificial neural network that captures the previous word information of a sequence in memory. The long short-term memory (LSTM) [37] network is a variant of RNN which learns long-term dependencies in a sequence by combining a gated memory unit and can avoid gradient disappearance and gradient explosion caused by RNN.

Mathematically, for each position t, the LSTM calculates $h_{t}$ using the current input vector $e_{t}$ and the previous state $h_{t-1}$:(3)$$f_{t}=\sigma \left(W_{f}e_{t}+U_{f}h_{t-1}+b_{f}\right)$$
(4)$$i_{t}=\sigma \left(W_{i}e_{t}+U_{i}h_{t-1}+b_{i}\right)$$
(5)$${\tilde{C}}_{t}=\tan h\left(W_{\tilde{C}}e_{t}+U_{\tilde{C}}h_{t-1}+b_{\tilde{C}}\right)$$
(6)$$C_{t}=f_{t}C_{t-1}+i_{t}{\tilde{C}}_{t}$$
(7)$$o_{t}=\sigma \left(W_{o}x_{t}+U_{o}h_{t-1}+b_{o}\right)$$
(8)$$h_{t}=o_{t}\tan h\left(C_{t}\right)$$
where $f,\;i,\;o\;\in \;R^{d_{h}}$ are the forget gate, input gate, and output gate, respectively ($d_{h}$ is a hyper-parameter indicating the dimensional of hidden state); σ is the element-wise sigmoid function; C is the cell vector; $W_{f},\;W_{i},\;W_{\tilde{C}},\;W_{o}$ are the weight matrices for hidden state $h_{t}$, input $e_{i}$, and memory cell $c_{t}$, respectively; $b_{f},\;b_{i},\;b_{\tilde{C}},\;b_{o}$ denote the bias vectors; tanh represents the hyperbolic tangent activation function; and $h_{t}$ is the output at time *t*.

The hidden state $h_{t}$ is obtained by LSTM only considering the previous state. We solve this problem with bidirectional LSTM, which considers both previous and future information. For any given input $e_{i}$ in Bi-LSTM, the final representation is:(9)$$h_{t}=\left({\vec{h}}_{t}⊕{\overset{\leftarrow }{h}}_{t}\right)$$

CRF Layer: For a sentence in the dataset, a word representation $h_{t}$ is obtained by the calculation of the previous method, considering the dependencies of adjacent markers. For example, the “I” (Inside) tag cannot be followed by “B” (Begin) or “E” (End) tag. Therefore, we use conditional random field (CRF) to predict a label sequence from a sequence of context representations, instead of using $h_{t}$ independently for tagging decisions.

For a given input sequence $X=\left[x_{1},\;x_{2},\;\ldots ,\;x_{n}\right]$, the output of the BiLSTM layer to input $x_{t}$ at time *t* is the feature vector representation $h_{t}$. The hidden layer output is mapped to the probability distribution vector $P=\left[p_{1},p_{2},\ldots ,p_{n}\right]$ on the label set using the softmax function after the BiLSTM output layer $P\epsilon R^{n\times k}$, where *k* is the number of different tags in the tag set and $p_{i,j}$ is the probability that the *i*th input is the *j*th tag. For a possible sequence of prediction results, $y=\left(y_{1},y_{2},\ldots ,y_{n}\right)$, where $y_{t}$ is the predictive label index at time *t*. We therefore define the score of the sequence as:(10)$$s\left(X,y\right)=\overset{n}{\underset{i=0}{\sum }}A_{y_{i},\;y_{i+1}}+\overset{n}{\underset{i=1}{\sum }}P_{i,\;y_{i}}$$
where A is called the transfer matrix and $A_{i,j}$ is the transition probability of tags *i* to *j*. This matrix is the parameter that needs to be trained in the CRF layer. It learns some important grammatical constraints that appear in the training corpus. Sequence y is actually a path chosen in chronological order in matrix P, and its probability of occurrence can be calculated by the softmax function:(11)$$p\left(y|X\right)=e^{s\left(X,y\right)}/\underset{\tilde{y}\in Y_{X}}{\sum }e^{s\left(X,\tilde{y}\right)}$$
where $Y_{x}$ is a set of observations of all possible predictive sequences for input sequence X. If, for instance, sequence y is the correct predictive result sequence, the logarithmic probability of the sequence needs to be maximized in the training process:(12)$$\log\left(p(y|X\right))=s\left(X,y\right)-\log(\underset{\tilde{y}\in Y_{x}}{\sum }e^{s\left(X,\tilde{y}\right)})$$

In the process of forecasting, we select the forecasting result $y^{*}$ to maximize its sequence score:(13)$$y^{*}=argmax\;s\left(X,\tilde{y}\right)\;\tilde{y}\in Y_{x}$$

### 3.3. Rules for Specific Task

In this section, two strategies for a particular task are presented. According to the particularity of the task, we need to find some of the entities from the data (note: not all entities here, e.g. these entities might be the primary or metastatic sites of cancer). For example, in ‘right lung cancer and right lung lower lobe inflammation’, ‘right lung’ is a primary sites entity and ‘right lung lower’ is not the entity required by the CHIP-2018 Task 1; thus, we cannot use the model to extract all the entities directly in the text. We use manual rules to resolve this problem. In addition, some of the entities shown in the data are incomplete, and the external domain dictionary is adopted to complete the entities or remove the excesses from the extracted entities.

Domain dictionary: We construct a dictionary of CEMR for entity of body parts referring to the training set (CCKS-2017 Task 2 and CHIP-2018 Task 1) and some open websites (e.g., “Baidu Baike”, Wikipedia, etc.). 

Rules: The purpose of defining rules is to identify sentences containing specific entities from the text. These sentences, referred to as candidate sentences, are divided into three categories: primary tumor site, lesion size, and tumor metastasis site. We define the rules in two aspects. (1) The statistical-based method extracts three types of sentences in the training data, and separately counts the three types to find the most frequent keywords in the sentence. (2) By observing and analyzing the data, we can obtain potential rules. For example, ‘lymph nodes’ and ‘mediates’ cannot be used as one entity alone. In the sentence ‘seeing a marked enlarged lymph node in the mediastinum’, the entity is ‘mediastinal lymph node’. Finally, we defined 12 rules. We list the six main rules in Table 3.

## 4. Results

### 4.1. Dataset and Experimental Settings

We used the CCKS-2017 Task 2 and CHIP-2018 Task 1 training datasets to conduct our experiments. The China Conference on Knowledge Graph and Semantic Computing is held by the Chinese Information Processing Society of China (CIPS), and it is the most famous conference about knowledge graph and semantic computing in China. The China Health Information Processing Conference (CHIP) is the annual seminar on medical, health, and biological information processing launched by the Technical Committee of the Chinese Information Processing Society (CIPS). In total, there are 1400 EMRs, of which CCKS-2017 has 800 (with five kinds of medical named entities, but we only selected sentences containing body parts as training sets) and CHIP-2018 has 600 (including two kinds of entities, namely body parts and lesion size). Each instance has one or more sentences. These sentences are further divided into clauses by periods or semicolons. The statistics of entities and sentence are listed in Table 4. We used 80% of the sentences as training data and the rest as test data.

We used TensorFlow to complete the CRF module. The parameters mainly include tag_indices and batch_size, where tag_indices represents the number of real tags and batch_size represents the number of samples processed in a batch. Furthermore, we also use many other parameters in our model. Parameter setting may influence the effect of the experiment. The parameter setting of the method proposed in this paper is shown in Table 5. 

### 4.2. Evaluation

The proposed method models the named entity recognition problem as a multi-classification problem. There are five categories in the CCKS-2017 dataset and three in the CHIP-2018 dataset. In the evaluation, they were converted into a two-category problem, and the specific conversion method is explained as follows.

According to the tagging pattern described in Section 4.1, each kind of entity recognition can be regarded as a five-category problem BIESO. We regard the complete recognition of a named entity as a correct recognition, and record it as the positive class of the two classifications, while only a part of the identified entity is regarded as an incorrect recognition, as the negative class, and the recognition result as O is also recorded as the negative class.

TP (True Positive) is the total number of entities matched with the entity in the labels. FP (False Positive) is the number of recognized labels that do not match the annotated corpus dataset. FN (False Negative) is the number of entity terms that do not match the predicted label entity.
(14)P=TP/(TP+FP)
(15)R=TP/(TP+FN)
(16)$$F=\frac{2*P*R}{P+R}$$
where P indicates precision measurement that defines the capability of a model to represent only related entities [38] and R (recall) computes the aptness to refer all corresponding entities. F is the harmonic average of P and R, which expresses the comprehensive effect of both.

### 4.3. Experimental Results

Our experiments were mainly divided into two parts. First, we carried out entity extraction model experiments on CCKS-2017 and CHIP-2018 datasets. Second, experiments were conducted for CHIP-2018-specific tasks.

In order to verify the effectiveness of the entity extraction model, we did several sets of experiments. Firstly, we compared with the basic Bi-LSTM-CRF model. Gridach [11], Habibi et al. [39], and Zeng et al. [40] successfully employed Bi-LSTM-CRF models with no additional features for English EMR. We compared the proposed (CWD-Bi-LSTM-CRF) model with the Char-Bi-LSTM-CRF models. We implemented five models with different features and various combinations. These models were test on two datasets (2017 CCKS Task 2 on body category and 2018 CHIP Task 1 on anatomic site.). All comparative results are summarized in Table 6.

Table 6 shows that the fifth model with char embedding, word embedding, and dictionary feature vector achieves the best performance: on 2018 CHIP Task 1 on anatomic site, it achieved a precision of 93.58%, a recall 94.20%, and a F1-score of 93.89%, while, on the 2017 CCKS Task 2 on body category, it achieved a precision of 91.38%, a recall 89.93%, and a F1-score of 90.65%. Because of the ambiguity of Chinese word boundaries, we found that word+Bi-LSTM model has the worst effect in all respects. Combining with char and word embedding, the F1 values of the two datasets were increased by 0.51% and 0.62%, respectively, compared with the first method; on CHIP dataset, the precision achieved the best results. In addition, the dictionary feature could bring benefit on the CHIP and CCKS datasets with improvement of 2.68% and 1.46% in term of F1-score, respectively. Finally, we tried the combination method of highway network and obtained the best result.

To investigate training speed, we compared our method with the basic Bi-LSTM-CRF in F1-score on the CHIP-2018 training dataset. As shown in Figure 4, our method converged at the fourth epoch, while the basic Bi-LSTM-CRF model did not converge until the eighth epoch. The convergence rate of the basic Bi-LSTM-CRF model is about twice that of our method. It is reasonable because our method includes dictionary vector features that facilitate the convergence of the model.

Besides the basic Bi-LSTM-CRF model, some existing methods incorporate additional features or approaches into Bi-LSTM-CRF models [29,30,31]. We also compared or method to a rule-based model and a machine learning model.

The comparative results are shown in Table 7. Because the corresponding reference method utilizes external resources that are not provided, we could not reproduce their model, thus some blank parts are reserved in the experiment results of CHIP-2018 dataset. Entities in both datasets can be considered as body parts, thus they can also be trained together. “Overall” indicates that both datasets were simultaneously used as training data. As shown in the table. The Bi-LSTM-CRF model achieved good results, increasing the F1 value by 0.63% compared to the model of Hu et al. [29]. Although the model of Hu et al. [30] had a better overall effect in the extraction of five types of entities, the F1 value being 91.08%, the entity extraction effect for the body category was not good. According to the evaluation results, our proposed model showed better performance on recognizing medical entity terms compared with the other models including CRF and rule-based model.

To further verify the experimental effect of the model on low-frequency entities and unknown entities, we first combined the 2017 CCKS Task 2 and 2018 CHIP Task 1 datasets. Then, we randomly selected 50% of the data as the training set, and the remaining 50% as the test set. The division was based on the number of times the entity appears in the training set. The criteria were: (1)Unknown entity test set: The entities in the sample have never appeared in the training set.(2)Low-frequency entity test set: The entities in the sample appear fewer than five times in the training set.(3)High-frequency entity test set: Entities in the sample appear more than five times in the training set.

As shown in Table 8, in the high-frequency entity test set, since the entities in the test set appear more times in the training set, the model could learn more features, thus all three algorithms achieved good results, and our method’s F1 score reached 98.08%. In the low-frequency entity and unknown entity test set, because there are too few samples for learning in the training set, fewer features could be learned. Compared with the other two methods, our method had the best results under these two types of test sets. Compared with Bi-LSTM-CRF, our method’s F1 score was improved by 30%, and our method was 20% higher than Bi-LSTM-Attention. Experiments proved that our method has greater advantages in low-frequency entities and unknown entity datasets.

Three entities were extracted from the CHIP-2018 dataset (600 training data and 200 test data), namely the primary site of the tumor, the size of the lesion, and the tumor metastasis site. The size of the lesion refers specifically to the size of the primary site of the tumor. Not all entities contained in EMRs belong to these entity categories, thus we only needed to extract some of the entities from EMRs. We used the rules to locate the candidate sentences of the three entities and obtained the entities from the sentences through the entity extraction method. The tumor origin site entity and tumor metastasis site entity could be directly obtained, while the size of the lesion needed to determine whether the primary site of the tumor is included in the sentence. In Table 9, we list the F1 values of the local test (using 10% of the training data as test data, and results obtained on a local computer) and the submitted results (using the test data given by the evaluation organization, and the results obtained on the evaluation organization’s server).

The method submitted to CHIP-2018 is to use rules to extract entity candidate sentences and char+Bi-LSTM-CRF model to extract entities. In the table, we can see that the local test and submission results are quite different: the gaps in F1 values are above 10%, and the gap in the metastasis site reached 23.36%. This shows that the generalization ability of the first method we submitted is not good.

We found that the previous method was not very effective on the CHIP-2018 task. Hence, we modified the method as: (1) Entity extraction is performed using the CWD-Bi-LSTM-CRF model with the same rules. (2) An external dictionary is used to correct the extracted entities. (3) The rules are redefined to extract candidate sentences. The experimental comparison results are shown in Table 9. The table uses the final criteria for the task, with weighted P, R, and F1. The weight of the primary site was 0.2, the weight of the lesion was 0.3, and the weight of the metastatic site was 0.5.

In Table 10, we can see that the F1 value was improved in the case of changing the entity extraction model and adding an external dictionary. The F1 value increased by 5.26% when we adopted the CWD-Bi-LSTM model, and it increased by 8.46% after we added the external dictionary. Finally, we redefined the rules, using 12 rules, and the F1 value reached 84.36%. The results demonstrate the necessity of adding rules and external dictionaries to the task.

## 5. Discussion

In this section, we discuss the proposed model in three aspects. First, we verify the robustness of the model, then analyze the advantages of the model under small data, and finally analyze the impact of the number of rules on the experimental results under specific tasks.

Robustness verification: Text is often noisy, e.g., including wrong words or less writing. Therefore, to verify the robustness of the model, we randomly extracted parts of sentences in the verification set, and randomly added, deleted or modified the words in the sentences. According to the proportion of modified sentences to all sentences in the verification set, we modified 1–10% of the sentences, respectively, and used the verification set in entity extraction model experiments for robustness verification. In Table 11, we can see that our model had strong stability. Compared with the experiment without noisy data, the F1 value did not fluctuate more than 1.17%. It shows that the noise in data has little effect on model efficiency. 

The advantage of our model under a small amount of data: We randomly selected 5%, 10%, 15%, 20%, and 25% data from the total dataset as the training dataset of the experiment. The experiment was conducted to compare the performance of our entity extraction model with the basic bidirectional LSTM-CRF model in terms of F1 scores. The experimental dataset used the body category classification of CCKS-2017 Task 2.

In Table 12, we can see that our model achieved 9.77% greater F1 value than the basic Bi-LSTM-CRF model when trained with 5% dataset. On the 10% dataset, the F1 value increased by 7.28%. For small-scale data, the average improvement rate was 4.91%. It demonstrates the benefits of dictionary features and character and word embedding for small-scale data. Moreover, these improvements decreased as the training dataset increased. This is because the more frequently entities occur in the training set and the fewer unknown entities there are, the better is the recognition performance that can be obtained from the basic model.

Effect of the number of rules: In the CHIP2018 evaluation task, entity extraction is required in conjunction with rules. In our experiment, however, we did not find that more rules lead better performance. In Figure 5, the abscissa represents the number of rules, and the ordinate represents the F1 value obtained by the experimental method. We can see that the increase in results depended on the importance of the rules instead of the number of rules. Because there is an influence relationship between rules, it is possible to add more rules that will result in lower results.

## 6. Conclusions

In this paper, we construct a new entity extraction model for CEMR by incorporating char, word embedding, and dictionary features. Compared to other popular entity extraction models, our model achieved better results when dealing with small frequency entities and unknown entities. In the CHIP-2018-specific task, we extracted entities by combining rules and domain dictionary. Through experiments, it was found that the introduction of rules and domain dictionary in the tasks could greatly improve the effect, thus proving the effectiveness of combining rules and domain dictionary in entity extraction tasks. In the future, we plan to continue to optimize our entity extraction model by introducing the attention mechanism. We will also study how to extract rules automatically. The amount of experimental data affects the experimental results to a certain extent. We will try to use the method of few-shot learning in the next step to solve the problem that EMR data are relatively small at present.

## Figures and Tables

**Figure 1 ijerph-17-02687-f001:**
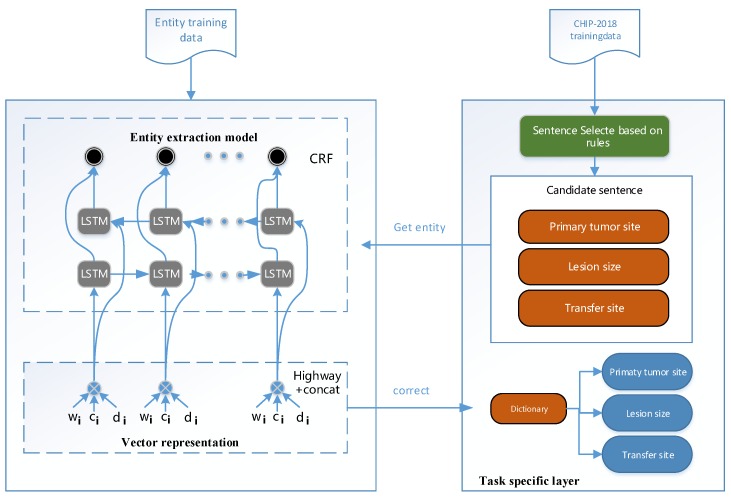
The general architecture of our proposed model. There are three parts in our model, namely vector representation, entity extraction model, and rules for specific task.

**Figure 2 ijerph-17-02687-f002:**
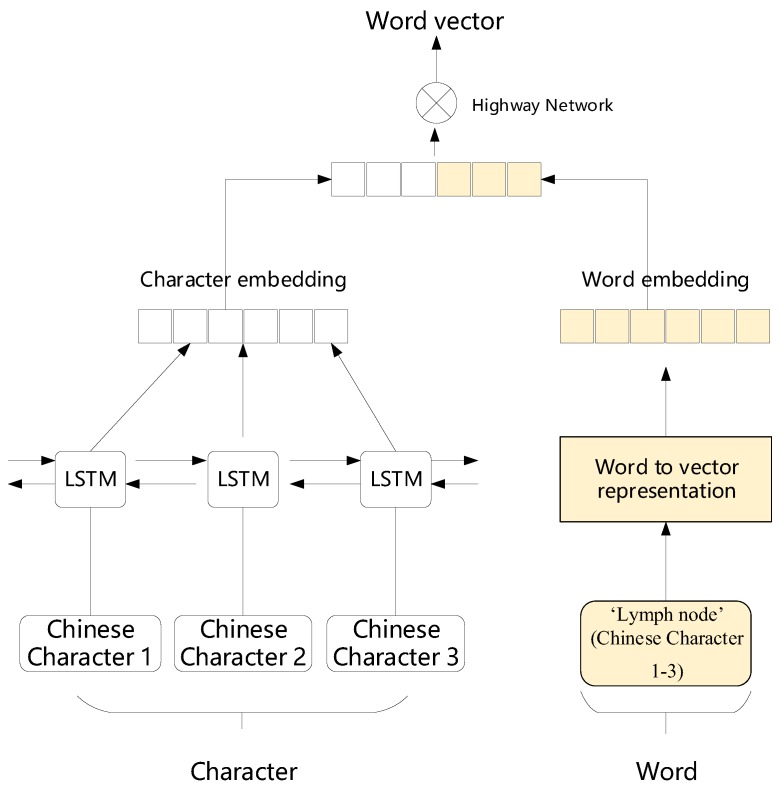
Word vector as combination of character embedding and word embedding. Each word trains a word vector. The word embedding training process is highlighted in color and the character embedding training process in white.

**Figure 3 ijerph-17-02687-f003:**
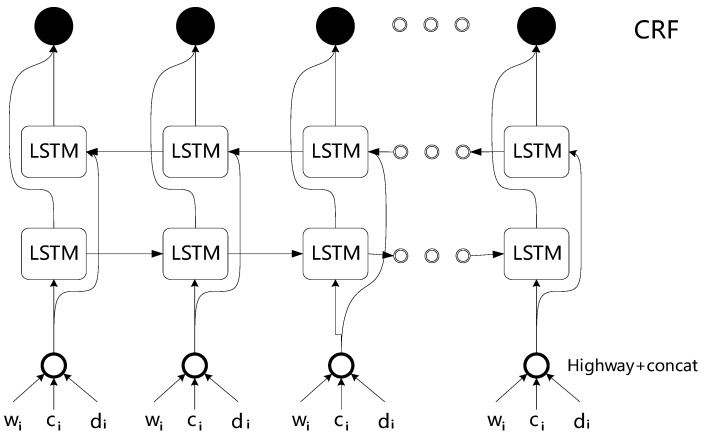
The main architecture of our entity extraction model.

**Figure 4 ijerph-17-02687-f004:**
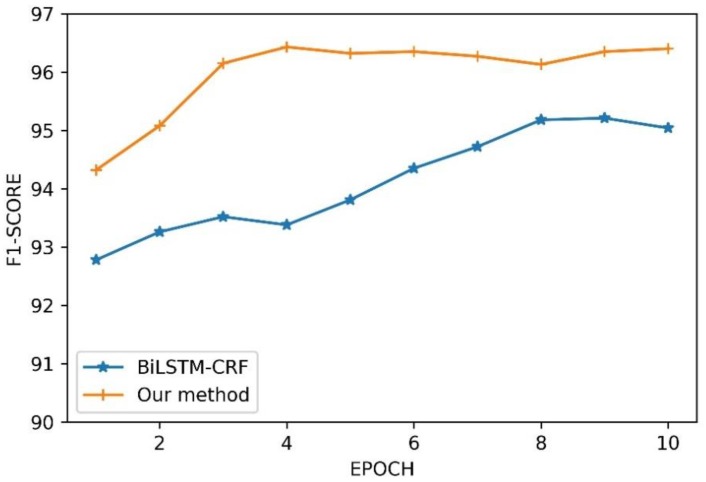
F1-score performance of different training epoch.

**Figure 5 ijerph-17-02687-f005:**
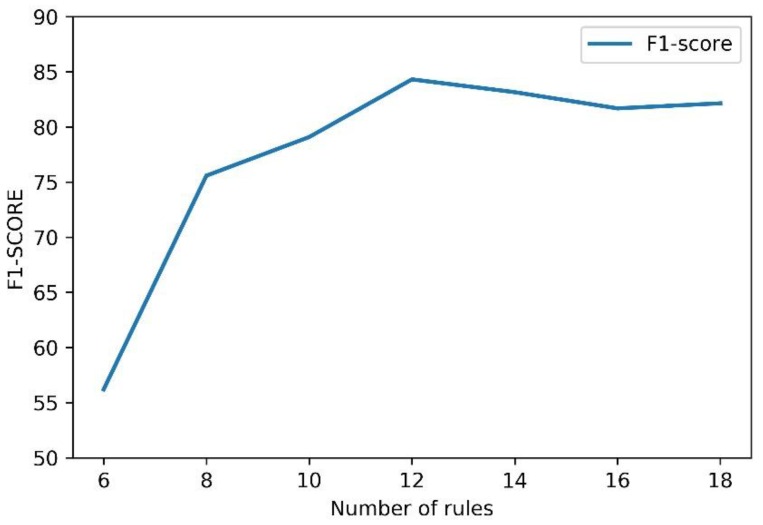
The effect of the number of rules on the results.

**Table 1 ijerph-17-02687-t001:** An example of the tag sequence.

Word Sequence	‘Right Side’	‘Internal Mammary’	‘Lymph Node’	‘Swollen’		‘Change’	‘Not’	‘Significant’	∘
Tag Sequence	B-b	I-b	E-b	O	O	O	O	O	O
Entity Type	Body	Body	Body	None	None	None	None	None	None

**Table 2 ijerph-17-02687-t002:** N-gram templates used to obtain text segments.

N-Gram	Templates
1-gram	$x_{i}$
2-gram	$x_{i-1}x_{i},\;x_{i}x_{i+1}$
3-gram	$x_{i-2}x_{i-1}x_{i},\;x_{i}x_{i+1}x_{i+2}$
4-gram	$x_{i-3}x_{i-2}x_{i-1}x_{i},\;x_{i}x_{i+1}x_{i+2}x_{i+3}$

**Table 3 ijerph-17-02687-t003:** Description of the six main rules.

Target	Rules
Sentence segmentation	Segment sentences by periods and semicolons; in special cases, the end of the sentence is unsigned, but the beginning of the sentence is numbered.
Candidate sentence for primary tumor site	Contains keywords: ‘cancer’, CA, MT.
Candidate sentence for lesion size	Contains keywords: (“cm” or “CM” or “MM”) and (‘density’ or ‘shadow’).
Candidate sentence for metastasis site	Contains keywords: ‘Metastasis’. If the sentence also contains the keyword of the original part of the tumor, the beginning of the candidate sentence is after the keyword.
Special case processing	Add lymph nodes after the part of entity (such as ‘Mediastinum’).
Primary site lesion size extraction	If the primary site of the tumor appears in the candidate sentence of the lesion size, the lesion size entity in the candidate sentence is extracted.

**Table 4 ijerph-17-02687-t004:** Statistics of entity and sentence.

Type	Number of Entities	Number of Sentences
CCKS-2017 body	8310	6523
CHIP-2018 body	13,124	5117
CHIP-2018 lesion size	1669	

**Table 5 ijerph-17-02687-t005:** Parameter setting of the proposed method.

Parameters	Value
Word vector embedding size	200
Dictionary feature vector embedding size	100
Number of hidden neurons for each hidden layer	300
batch_size	64
tag_indices	4
Learning rate	0.005
Number of epochs	10
Dropout out	0.5
Optimizer	Adam optimizer

**Table 6 ijerph-17-02687-t006:** Comparative results of four different feature combination models.

Methods	CHIP-2018 Task 1 on Anatomic Site			CCKS-2017 Task 2 on Body Category		
	P	R	F1	P	R	F
Char+Bi-LSTM	93.26	89.63	91.41	89.13	87.01	88.05
Word+Bi-LSTM	91.76	86.92	89.27	85.47	83.68	84.57
Char+word+Bi-LSTM	**93.69**	90.22	91.92	89.52	87.83	88.67
Char+word+dict+Bi-LSTM	93.31	93.89	93.60	90.68	89.59	90.13
Char+word+dict+Bi-LSTM(Highway+concat)	93.58	**94.20**	**93.89**	**91.38**	**89.93**	**90.65**

In order to show the comparison in Table 6 more intuitively, the best experiment results are in bold.

**Table 7 ijerph-17-02687-t007:** Comparative results between state-of-the-art models and our method.

	Dataset	2017 CCKS Task 2 on Body Category	2018 CHIP Task 1 on Anatomic Site	Overall
Methods				
Rule-based		82.32	*	*
CRF		86.89	88.62	87.34
Bi-LSTM-CRF		88.05	91.41	89.73
Bi-LSTM-CRF-N-F [29]		85.77	*	*
Vote [30]		87.42	*	*
Bi-LSTM-Attention [31]		89.21	92.37	91.46
Our method		**90.65**	**93.89**	**93.13**

In order to show the comparison in Table 7 more intuitively, the best experiment results are in bold. *: reserv in the experiment results of CHIP-2018 dataset.

**Table 8 ijerph-17-02687-t008:** F1 score of each method under different data types.

	Dataset	Unknown Entity Test Set	Low-Frequency Entity Test Set	High-Frequency Entity Test Set
Methods				
Bi-LSTM-CRF		39.65	56.83	91.13
Bi-LSTM-Attention [31]		46.32	64.71	94.52
Our method		**74.33**	**86.89**	**98.08**

In order to show the comparison in Table 8 more intuitively, the best experiment results are in bold.

**Table 9 ijerph-17-02687-t009:** F1-scores of first submitted methods on each category.

Test Method	Primary Site	Lesion Size	Metastasis Site
Local test	76.92	78.74	85.18
Submitted	65.49	62.84	61.82

**Table 10 ijerph-17-02687-t010:** Results of various methods on CHIP-2018 test set.

Method	P	R	F1
Frist submit	62.73	61.90	62.31
CWD-Bi-LSTM+rule	67.84	67.32	67.57
CWD-Bi-LSTM+rule +dictionary	76.38	75.69	76.03
CWD-Bi-LSTM+rule* +dictionary	**85.26**	**83.49**	**84.36**

**Table 11 ijerph-17-02687-t011:** Model robustness verification.

Noise Ratio	0%	1%	2%	3%	4%	5%	6%	7%	8%	9%	10%
F1-score	93.13	92.89	92.67	92.86	92.25	91.96	92.61	92.39	92.01	92.49	92.28

**Table 12 ijerph-17-02687-t012:** Comparison of results under different datasets.

	%Dataset	5%	10%	15%	20%	25%
Methods						
Bi-LSTM + CRF		60.68	70.28	78.52	83.26	85.48
Our method		70.45	77.62	82.47	85.38	86.93

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
