# Peer review of "Improving the Named Entity Recognition of Chinese Electronic Medical Records by Combining Domain Dictionary and Rules"

_ijerph, 2020, doi:10.3390/ijerph17082687_

Round 1

Reviewer 1 Report

My comments:

1) Additional experiments have to be carried out to validate the feasibility and robustness of the proposed system.

2) There is no notation table to understand what math symbols stand for? Adding the notation table is necessary.

3) It is difficult to follow up on the formula! Some of them are the right position other ones in left. They are not justified!

4) Line 134: Error! Reference source not found. Proofreading. I suggest to the authors a careful re-reading of the manuscript, rewriting some sentences in a better way and subdividing some long parts of the text into several paragraphs, to improve the readability.

5) The paper could improve in the state of the art. Why didn’t consider the blockchain perspectives? I suggest considering the following papers:

https://www.mdpi.com/2076-3417/9/9/1736

https://www.sciencedirect.com/science/article/pii/S0065245818300196

Author Response

Thank you for your valuable advices. We have modified our manuscript according to your comments. Please see the attachment.

Reviewer 2 Report

The manuscript is well written and structured.
It presents a neural network model for the detection of medical entities in Chinese.
But there are things that can be improved:
- In Introduction Section you say that the state-of-the-art is BiLSTM-CRF, are you sure? http://nlpprogress.com/english/named_entity_recognition.html
https://paperswithcode.com/sota/named-entity-recognition-ner-on-conll-2003
- In related work you have not mentioned any significant challenges at NLP conferences.
- In related work you have not mentioned the novel methods of transfer learning: BERT, ELECTRA, T5, among others. These methods have achieved really good results in the domain of NER.
- In the CRF layer, what parameters did you use?
- No error analysis has been conducted.
- A study of the most relevant rules has not been presented.
- The word embeddings used have not been introduced.

Regarding the style of the manuscript, there are some problems:
- There's an unopened bracket on line 99: 4,11,12]
- Some references are bracketed and some are not. I recommend you use for example: [34]
- There are errors in references to tables or images: lines 134, 167 and 202.
- I recommend extending the width of table 1.
- I recommend extending the width of: table 1 and the first column of table 2.
- It is difficult to read the equations, they use several lines in the PDF.
- In table 4, is the number of sentences missing from any corpus? It is not well defined.
- There are places in the manuscript where space or commas are missing, for example, line 106: (SVMs) 1415, Maximum...; line 165: by Mikolov22.; line 337: 93.89%, in 2017ccks-task2 on body... Please review the entire article carefully.

Author Response

(The authors gave the same response as above.)

Reviewer 3 Report

In this manuscript, an effort has been made to propose a hybrid entity extraction model to extract entities from CEMR. Experiments were conducted and results were shown proving the improvement in performance. However, the manuscript needs major revisions before consideration of publication.

Comment 1: The structure of the manuscript needs major revisions. In the Related Work, the scope should be how previous studies or efforts have been done to specifically target the NER difficulties in the Chinese language, rather than general reviews of different methods. They might not necessarily be in the medical/clinical domain. However, it would be more relevant to the scope of this study compared with NER methods targeted for English. In the Introduction, the author mentioned the challenges and difficulties that NER faced in the medical or clinical domain and specifically pointed out that it is more difficult for the Chinese language. Detailed descriptions should be expected. 

The author did mentions some other methods proposed by other researches that were targeted for Chinese, in around Lines 354-362. However, these contents should be addressed earlier in the paper rather than in the Results section. 

Comment 2: The author claimed that the proposed method has a good effect on dealing with small frequency entities and unknown entities. It was listed as one of the main contributions of this study. However, the relevant results were only shown in the Discussion section. And there were no experiments and results shown on the bigger frequency dataset. More experiments should be conducted on a larger proportion of the dataset and results should be shown and analyzed in the Results section. 

Comment 3: The scale of the F1-score in Fig. 1 and Fig. 2 should be consistent. It should be either the number out of 100 or in percentage. 

Comment 4: The use of language needs extensive editing for this manuscript. There exist obvious typos and errors even at the beginning of the manuscript. Such as Line 38: "Specifically, In order to" and Line 57: "in EMR are nosier" I assumed the author meant "noisier".

Comment 5: The references in the manuscript need to be carefully examined and re-ordered. Some references have square brackets while some don't. Besides, the reference order is disorganized. For example, Line 44, reference 8 appeared while there was not even reference 2. Another example, Line 134, 167, 202,, the error message is clearly shown. 

Comment 6: Most equations have an error in formatting, which makes them difficult to read. Lines: 214-221.

Author Response

(The authors gave the same response as above.)

Round 2

Reviewer 1 Report

All is good. The paper is technically sound and provides novel ideas to me.